# Trends in the Prevalence and Case Characteristics of Child Sexual Abuse in Mexico, 2018–2023

**DOI:** 10.3390/healthcare13192489

**Published:** 2025-09-30

**Authors:** Leonor Rivera-Rivera, Marina Séris-Martínez, Paola Adanari Ortega-Ceballos, Arturo Reding-Bernal, Claudia I. Astudillo-García, Lorena Elizabeth Castillo Castillo, Luz Myriam Reynales-Shigematsu

**Affiliations:** 1Centro de Investigación en Salud Poblacional, Instituto Nacional de Salud Pública, Cuernavaca C.P. 62100, Mexico; lrivera@insp.mx (L.R.-R.); dra.seris.martinez@gmail.com (M.S.-M.); lreynales@insp.mx (L.M.R.-S.); 2Facultad de Enfermería, Universidad Autónoma del Estado de Morelos, Cuernavaca C.P. 62209, Mexico; 3Dirección de Investigación, Hospital General de México “Dr. Eduardo Liceaga”, Ciudad de Mexico C.P. 06720, Mexico; reding_79@yahoo.com; 4Dirección de Investigaciones Epidemiológicas y Psicosociales, Instituto Nacional de Psiquiatría Ramón de la Fuente Muñiz, Ciudad de Mexico C.P. 14370, Mexico; claudiaiveth.astudillo@gmail.com; 5Instituto de la Mujer de Cuernavaca, Ayuntamiento de Cuernavaca, Cuernavaca C.P. 62170, Mexico; imujer@cuernavaca.gob.mx

**Keywords:** child sexual abuse, prevalence, trend, Mexico

## Abstract

**Background**: Child sexual abuse (CSA) is a serious public health concern that violates the rights of children. In Mexico, little is known about the actual figures for this type of violence. **Objective**: This study aimed to determine trends in the prevalence and case characteristics of CSA in a representative sample of children in Mexico. **Materials and Methods**: Data from the National Health and Nutrition Survey (ENSANUT) for 2018, 2020, 2021, 2022 and 2023 were used (n = 24,179). Proportions of CSA were estimated using the weighted mean of a binary variable, and the variance of the estimated proportion was calculated using the Taylor linearization method. Logistic regression models were estimated, and Adjusted Odds Ratios (AORs) with 95% Confidence Intervals (95% CIs) were obtained. **Results**: The prevalence of CSA ranged from 2.22% (2018) to 5.66% (2023). There was an increasing trend in CSA between 2018 and 2021, which was even more pronounced (154.95%) between 2018 and 2023 (*p* < 0.001). The main perpetrator in CSA cases was a family member (78.51%), and most victims did not report the abuse to the authorities. Girls were more likely to experience CSA (AOR = 2.83, 95% CI: 1.72–4.68), and as years passed (from 2018 to 2023), the likelihood of becoming a victim of CSA increased. **Conclusions**: CSA is a problem that has increased in recent years in Mexico. It is noteworthy that the main perpetrator is within the family, which may influence the lack of reporting of these cases. In view of this situation, it is necessary to implement strategies to prevent CSA in children, involving mothers, fathers, and caregivers.

## 1. Introduction

Child sexual abuse (CSA) is a serious public health concern that violates the rights of children [1], robs them of their dignity, and affects their physical and mental health [2]. According to the Convention on the Rights of the Child, a child is defined as any person under the age of 18 years [1]. The World Health Organization includes these ages in its definition of CSA: “*the involvement of a child in a sexual activity that he or she does not fully comprehend, is unable to give informed consent to, or for which the child is not developmentally prepared and cannot give consent, or that violates the laws or taboos of a society*” [3].

Globally, it is estimated that 20% of girls and 8% of boys have suffered from CSA [4]. A meta-analysis on the global prevalence of CSA in Latin America found a prevalence of 13.4% (95% CI: 6.2–26.5) in women and 13.8% (95% CI: 3.7–40.0) in men [4].

Cagney et al. [5] employed spatiotemporal Gaussian process regression to estimate the complete time series of exposure to sexual violence against children across different age-sex-country combinations. Their findings revealed the global age-standardized prevalence in 2023 was 18.9% (with a 95% uncertainty interval [UI] of 16.0 to 25.2) for females and 14.8% (95% UI of 9.5 to 23.5) for males. In Latin America and the Caribbean, the estimated prevalence was 17.6% (95% UI of 14.9 to 20.9) for women and 13.9% (95% UI of 6.8 to 25.4) for men. In Mexico, the prevalence figures were 17.4% (95% UI of 9.2 to 27.7) for women and 13.6% (95% UI of 5.6 to 27.7) for men, which were closely aligned with the global estimates.

Piolanti et al. [6] conducted a meta-analysis to estimate the global prevalence of sexual violence against children by examining population-based studies at the country level. They reviewed 165 studies involving a total of 958,182 children from 80 countries, with a particular focus on data concerning girls. The findings revealed that lifetime sexual harassment was the most frequently reported outcome, with a combined rate of 11.4% (95% CI: 8.5–15.1%). Additionally, the rates of lifetime completed forced sexual intercourse were higher among girls (6.8% [95% CI: 6.1–7.6%]) compared to boys (3.3% [95% CI: 2.5–4.3%]). In line with findings from Cagney et al., this study indicates considerable variation in the reported prevalence of sexual violence across different regions, countries, and between girls and boys.

In Mexico, little is known about the actual figures for this type of violence, with most information obtained retrospectively through CSA reports, mainly from adolescent and adult women [7]. According to data from the 2021 National Survey on the Dynamics of Household Relationships (ENDIREH), 12.6% of women aged 15 and older experienced sexual abuse in childhood, showing an increase compared to 2016 (9.4%) [8]. Data from the 2018–19 National Health and Nutrition Survey (ENSANUT) showed a national prevalence of CSA in the population aged 10 to 19 years of 2.5%; 3.8% in women and 1.2% in men [9].

CSA is a problem that is rarely reported [10], mainly due to fear of reprisals by the perpetrator, who in most cases is a family member or someone close to the victim [5,7,9], leading to underreporting of this phenomenon [11,12,13]. Recognizing that a child has been a victim of CSA often involves questioning religious beliefs, social norms, and family values, which, more often than not, leads to keeping it a family secret [14].

The safeguarding of children from sexual abuse in Mexico relies on a comprehensive legal framework that includes provisions from the Constitution, specific laws, and international commitments. In this context, Article 4 of the Political Constitution [15] establishes that: “*In all decisions and actions of the State, the principle of the best interests of the child shall be safeguarded and complied with, fully guaranteeing their rights. Boys and girls have the right to the satisfaction of their needs for food, health, education, and healthy recreation for their integral development. This principle shall guide the design, execution, monitoring, and evaluation of public policies directed at children. Parents, guardians, and custodians have the obligation to preserve and demand compliance with these rights and principles. The State shall provide facilities to individuals to assist in the fulfillment of children’s rights*.”

This mandate is implemented through the National Code of Criminal Procedure [16], stresses in Article 222 that “*It is the obligation of any person who has knowledge of cases of children and adolescents who suffer or have suffered, in any way, a violation of their rights, to immediately bring it to the attention of the competent authorities*…”. Furthermore, Article 47 states that “The federal, State, and municipal authorities… within the scope of their respective powers, are required to take the necessary measures to prevent, address, and punish cases in which children or adolescents are affected by: I. Neglect, negligence, abandonment, or physical, psychological, or sexual abuse;…”.

The General Law on the Rights of Girls, Boys, and Adolescents (LGDNNA) [17] establishes a comprehensive framework for the protection and action provided by the Protection Attorney’s Offices and the National System for the Comprehensive Protection of Girls, Boys, and Adolescents (SIPINNA). This system is a collection of institutions and mechanisms in Mexico that are responsible for developing and implementing public policies aimed at guaranteeing the rights and protection of individuals aged 0 to 17. The law’s primary objective is to ensure that these rights are upheld, allowing girls, boys, and adolescents to live in environments free from violence. It also emphasizes the responsibility of authorities to prevent, address, and manage cases of violence and abuse.

In addition, the Mexican Official Standard (NOM)-046-SSA2-2005 [18] stipulates that health personnel are required to promptly report any suspected cases of sexual violence to the Public Ministry, especially when it involves minors. According to the NOM Section 6.5.5, “*If the affected individual is a minor or unable to make decisions legally, the appropriate law enforcement agency must also be notified*”.

In the criminal justice system, the Federal Penal Code (articles 202, 260, 261, and 262) [19] and the penal codes of Federal States impose severe penalties for crimes such as sexual abuse, rape, harassment, child pornography, and commercial sexual exploitation. These penalties are particularly heightened when the victim is under 18 years old. Furthermore, the General Victims Law [20] aims to protect the rights of victims in cases of crimes and human rights violations, requiring authorities to provide assistance, support, and comprehensive reparations.

To comply with current legislation and ensure the protection of children, we require up-to-date data on the prevalence of child sexual abuse (CSA), especially from representative samples. This issue often occurs within the family—where children should feel safe and cared for—which makes it difficult to prevent, identify, and address. Therefore, a rigorous analysis and detailed breakdown of the data are essential. This will enable different government sectors, civil society organizations, and the community as a whole to make a more strategic impact in eradicating this serious problem in Mexico. The objective of this study was to analyze trends in CSA prevalence and characterize reported cases in a representative sample of children in Mexico, to generate evidence that contributes to the design of public policies aimed at the prevention, detection, and management of CSA in the country.

## 2. Materials and Methods

### 2.1. Data

The National Health and Nutrition Survey (ENSANUT) is Mexico’s primary epidemiological research tool, providing essential data on the health and nutritional status of the Mexican population across various age groups, including children under five, adolescents, adults, and older adults. Importantly, ENSANUT also addresses critical indicators of child sexual violence, making it a vital data resource for understanding public health challenges. Originally, ENSANUT was conducted every five years—in 2006, 2012, 2016 (intermediate wave), and 2018. However, since 2020, ENSANUT has adopted a new approach, transitioning to an annual survey to better capture the evolving landscape of public health needs. These surveys employ a probability-based, stratified, and cluster sampling design to ensure national representativeness [21].

For this analysis, we collected data from a sample of 24,179 individuals aged 10 to 17 years, with the following breakdown by year: 2018 (14,575), 2020 (1749), 2021 (3372), 2022 (2888), and 2023 (1592).

To select participants for the study, households were randomly chosen. Once selected, the number of household members was listed, and one child was randomly selected to complete a health questionnaire. Informed assent was obtained from the children, along with consent from their parents or legal guardians. All information collected is anonymous. Importantly, the entire field team, including interviewers and supervisors, is trained and standardized in administering health surveys to ensure the confidentiality and quality of the data collected. ENSANUT is conducted using electronic devices, specifically tablets. For sensitive topics—such as substance use, mental health, and violence—children were able to answer questions on the tablets in a private space within their homes, which helped maintain confidentiality. We performed all analyses using Stata v. 16 (College Station, TX, USA) [22]. The survey protocols received approval from the Ethics, Research, and Biosafety Committees of the National Institute of Public Health, with the following (CI 2018: 1557; CI 2020:1679; CI 2021: 1750; CI 2022: 1807; CI 2023: 1865).

### 2.2. Study Variables

#### 2.2.1. Sociodemographic Variables

The variables used were gender (male and female), age (10–13 years, 14–17 years), place of residence (rural: towns with <2500 inhabitants and urban: towns with ≥2500 inhabitants) and socioeconomic status (SES), divided into high, medium, and low.

#### 2.2.2. CSA Variables

##### Child Sexual Abuse

The CSA variable was based on the question used in all surveys: “Throughout your life, has anyone ever fondled, touched, or caressed any part of your body or had sex with you when you were very young?” This question had the following response options: Yes, before age 12; Yes, when I was 12 or older; and No, never. Those who answered yes to the question were considered CSA cases, regardless of the age when it occurred.

##### Gender of the Perpetrator

The gender of the perpetrator was determined by asking the question, “Was the person who did it a man or a woman?”

##### Relationship with the Perpetrator

To determine the relationship with the perpetrator, the question “What was your relationship with that person?” was asked. The possible answers were: partner; family member; friend; boyfriend/girlfriend; neighbor; and stranger.

##### CSA Treatment

To determine whether the victim received treatment after the assault, the following question was asked: “Who treated you after the attack?” Multiple response options were provided (no one, clinic, psychologist, bone setter, traditional healer, etc.). This variable was combined and dichotomized, resulting in 0 = No one, 1 = Received treatment.

##### CSA Reporting

The following question was asked: “Did you or your family report the person who assaulted you to the authorities?” With response options 0 = No and 1 = Yes.

##### Reason for Not Reporting CSA

Those who answered that they had not filed a report were asked, “Why didn’t you report it?” The options for answering this question were: fear; shame; threats; I didn’t know I could; and other.

##### CSA Reporting Location

To determine where reports were filed, the following question was used: “What authority did you report it to?” (Public Prosecutor’s Office, DIF, trustee, other).

### 2.3. Statistical Analysis

To conduct the statistical analysis, we used the complex sample design of the 2018, 2020, 2021, 2022, and 2023 National Health and Nutrition Surveys, which includes the use of weights, strata, and primary sampling units, all of which are nationally representative. The variables of interest (per survey) were explored, obtaining proportions with 95% confidence intervals (95%CIs). The prevalence of CSA was estimated by survey year and sociodemographic characteristics. The estimate of CSA proportions with their corresponding confidence intervals per year was obtained using a weighted mean of a binary variable, and to calculate the variance of this estimated proportion, the Taylor linearization method was used, which takes into account the stratified design, the primary sampling units, and the sample weights specific to ENSANUT.

Multiple logistic regression models were estimated, obtaining Adjusted Odds Ratios (AORs) by survey year, sex, age, residence, and socioeconomic status with their 95% CI using complex sample design. Since there are five surveys from different years and each one covers approximately the same population size, the weight was divided by five, covering 18,017,388 children aged 10 to 17 years in Mexico. This adjusted weight, as well as the sample design, is used in all estimates presented in this manuscript. The Hosmer-Lemeshow goodness-of-fit test, ROC curves, and the area under the curve were used to diagnose the final multiple logistic regression model. Analyses were performed using the Stata 17 statistical package [22].

## 3. Results

### 3.1. Description of the Study Population

The number of boys was higher than the number of girls in each of the surveys; the predominant age group in 2018, 2020, and 2023 was 14 to 17 years old, while in 2021 and 2022, the predominant age group was 10 to 13 years old. Approximately 75% of the population was concentrated in urban areas and was evenly distributed across the three socioeconomic levels for all five surveys (Table 1). Most children who suffered CSA reported that the abuser was male (86.29% to 94.93%). In the five surveys, in almost half of the cases or even a higher percentage, the abuser was a family member (43.13% to 78.51%). Between 14.29% and 30.62% of those who suffered CSA received treatment after the assault, while between 6.69% and 19.52% reported the crime to the authorities; fear was the main reason for not reporting the crime in all five surveys. The Public Prosecutor’s Office was the institution where between 60.74% and 91.65% of children filed their reports (Table 1).

### 3.2. Prevalence of CSA by Sociodemographic Characteristics and Survey Year

The overall prevalence of CSA ranged from 2.22% to 5.66% between 2018 and 2023. Figure 1 also shows the prevalence rates by survey year, as well as their corresponding confidence intervals. CSA occurred in both sexes, but was more prevalent among girls, showing a gradual increase throughout the different surveys (from 3.32% to 7.08%). The population aged 14 to 17 years reported a higher prevalence of CSA in the different surveys. An increase in CSA was observed in both rural and urban areas, although the prevalence was higher in urban areas (2.44% in 2018 and 6.27% in 2023). In terms of SES, in 2018, 2021, 2022, and 2023, the prevalence of CSA was higher in the middle range (2.43%, 3.44%, 5.33%, and 7.77%, respectively), while in 2020, this prevalence was higher in the high range with 3.80% (Table 2).

### 3.3. Sociodemographic Factors Associated with CSA

The results of the multiple logistic regression model showed that as the years pass (2018–2023), the likelihood of CSA increases significantly. Likewise, girls were almost three times more likely to be victims of CSA than boys (AOR = 2.83, 95% CI 1.72–4.68). In addition, individuals aged 14 to 17 were almost twice as likely to experience CSA compared to those aged 10 to 13 (AOR = 1.89, 95% CI 1.25–2.86). SES and place of residence were not significantly associated with CSA. (Table 3).

## 4. Discussion

The results of this study, based on representative samples of the population of minors under 18 years of age in Mexico between 2018 and 2023, show an upward trend in the prevalence of CSA at the national level, with girls being at a higher risk than boys. In addition, some conditions and sociodemographic factors associated with such violence in childhood were found.

In this study, the increase in the prevalence of child sexual abuse temporarily coincides with the peak period of the COVID-19 pandemic and the post-pandemic period. The increase was particularly marked in 2023, as girls and boys were almost three times more likely to be victims of CSA compared to 2018.

These findings are similar to those of other studies, which have also reported an increase in the prevalence of CSA after the pandemic, both internationally [23,24] and in Mexico [25]. A study conducted in China found that before the lockdown caused by the COVID-19 pandemic, the prevalence of CSA was 1.6%, while after the lockdown was lifted, CSA increased significantly to 2.9% (*p* = 0.002) [26]. In Mexico, a study conducted in Mexico City found a progressive increase in CSA during the COVID-19 pandemic [25].

The findings indicate that confinement conditions during the pandemic increased the risk of child sexual abuse (CSA). Many children spent more time at home with potential abusers, often family members. Additionally, the rise in unemployment and poverty experienced by the Mexican population during this sanitary crisis created greater family stress, leading to an increase in domestic violence and alcohol consumption, both of which are recognized risk factors for CSA [27]. Furthermore, a culture of impunity and underreporting of CSA persists in Mexico, allowing offenders to continue their crimes without consequence. The increased use of social media during the COVID-19 pandemic also heightened the risk of sexual violence, including digital abuse, against children [28].

Although our study indicated an increase in the prevalence of Child Sexual Abuse (CSA), these estimates are lower than those reported in other research studies [4,5,6]. We believe this underestimation may be due to the sensitive nature of the issue. Children who experience CSA often feel shame, guilt, and fear toward the perpetrator, especially when the aggressor is a family member [29]. Additionally, CSA remains a taboo topic in Mexico, and disclosing such experiences can lead to stigmatization, causing many children to withhold this information during surveys. There may also be memory bias or psychological denial mechanisms at play, as individuals may struggle to confront the trauma associated with sexual violence [30].

It is noteworthy that the ENSANUT includes only one question to measure child sexual abuse, while most other studies typically incorporate multiple items for a more comprehensive assessment. Furthermore, this survey is primarily designed to investigate major health problems in the Mexican population and does not explicitly focus on child sexual abuse. Therefore, the findings related to child sexual abuse provide a general overview of the situation and seek to draw the attention of decision-makers. It is critical to recognize that currently, there is no other nationally representative survey in Mexico that analyzes child sexual abuse among both women and men.

Consistent with our results, international [31,32] and Mexican studies [25] have found that CSA is more prevalent in girls than in boys, and although boys are not exempt from this problem, girls were more likely to experience CSA (AOR = 2.83, 95% CI 1.72–4.68), highlighting the issue of inequality in terms of gender-based violence, which is very present in the Mexican context [33]. They also reveal cultural and family practices that perpetuate roles of subordination and silence. ENSANUT, in its original design, only includes participants’ sex as a variable, omitting gender identity. It is essential to consider including gender identity as a variable in future data collections to gain a more comprehensive understanding of ASI among non-hegemonic gender identities.

In terms of age, our study found that people aged 14 to 17 were more likely to be victims of CSA than those aged 10 to 13. These results are consistent with studies conducted in Mexico [19] and Brazil [34]. In the latter study, a higher prevalence was observed in participants aged 14 to 18 than in participants aged 10 to 13 [34]. However, a study conducted in the United States reports data that contradict ours, showing that the probability of CSA decreased with each year of age [31]. These differences could be due to the prevailing family dynamics in each country, the degree of access to child protection systems, and the aggressors’ profiles.

Contrary to what one might believe, CSA is a problem that occurs regardless of place of residence (rural or urban) and socioeconomic status. Although it may be more frequent in specific population groups (due to higher risk factors), any child can be a victim of CSA, irrespective of socioeconomic status or place of residence. These findings are consistent with research conducted in Brazil, which found no difference in CSA by SES [34]. Additionally, the study by Long et al. also found no association between household income or type of area (urban) and CSA [26].

Our results showed that the main perpetrator of CSA is a man, and in most cases a family member. These findings are consistent with other studies, such as that by Oliveira et al. [34], who found that the main perpetrator was a man (82.4%) from within the family (64%).

Children and adolescents who are victims of sexual assault generally face traumatic situations both during and after the assault, for which they cannot find an explanation. They are then torn between anguish, doubts, uncertainties, and often conflicting emotions. This becomes even more intense when the abuser is a family member with whom they had a positive emotional relationship prior to the abuse [35], leaving them unsure of how to act or prevent it from happening again, unable to understand their feelings, fears, or their relationship with their abuser at home [36]. Authors such as Díaz [37], Losada and Jursza [38], and Pérez, Ordoñez, and Amador [39] agree that CSA occurs mainly through an abuse of power, inequality arising from age, authority, role in the family, and the vulnerability of the victim, who in many cases ends up not disclosing the violent incident due to threats, lies, blame, blackmail, and psychological manipulation by the abuser [40].

On the other hand, CSA, being a problem that occurs mainly within the family unit, can be considered a taboo subject and often involves questioning religious and social beliefs, as well as family values, resulting in a family secret [14] being created when a child discloses the abuse. And so, even if CSA is disclosed to a caregiver, families may dismiss the story of sexual abuse, giving greater credibility to the adult for economic, social, moral, and other reasons [34]. This is why the protective role of the family is most concerning when children are abused by those who should defend them the most [36].

The low reporting rate is one of the most problematic issues when it comes to the criminal justice system’s response to child sexual abuse. On top of that, many victims have negative and traumatic experiences when they come into contact with different agents in the criminal justice system. Children and young people often choose not to disclose sexual abuse, which prevents them from getting help and allows perpetrators to continue undetected. When CSA victims decide to come forward, they face a number of barriers, such as limited support, perceived negative consequences, feelings of guilt or shame.

It is important to highlight that, although Mexico has a legal framework that provides for the protection of children, as well as agencies that prevent and respond to violence against this age group, these are still not well-developed enough to fully address CSA cases in the country [41], as they work in isolation. It is therefore essential to promote coordinated and inter-institutional collaboration between the different agencies working in both CSA prevention and treatment, including civil society, educational institutions, health institutions, security institutions, and other related bodies.

This study is not without limitations; as it is a cross-sectional study, causality cannot be established. However, the results are internally valid as they are based on a random sample representative of the child population of Mexico at the national and regional levels, and the results are consistent with previous studies.

Although ENSANUT is not violence-specific survey, it still has limitations in assessing various aspects of child sexual abuse (CSA). This includes gathering information about the characteristics of both the victims and the abusers, as well as the different types of CSA. It is important to note that the ENSANUT was initially conducted every five years to align with the start of each government administration. The most recent edition conducted under this scheme was ENSANUT 2018. However, since 2020, it has become necessary to monitor the evolution of public health issues in the context of the COVID-19 pandemic. For this reason, the survey is now conducted on an annual basis [21]. The sample sizes have been established to be representative at the national level and to allow for regional and state representativeness over the period. However, despite these limitations, ENSANUT is valuable because it helps identify health problems at the population level, including CSA, and raises awareness among decision-makers. In contrast to cohort studies or intervention trials that follow individual cases over time, the ENSANUT is a cross-sectional survey that maintains anonymity by not identifying specific individuals. As a result, neither the field staff nor the data analysts can make specific diagnoses of health issues, so children are not referred to particular health services.

For future research, it is important to generate and separate the questions used to measure CSA, as it includes diverse acts of child sexual violence.

## 5. Conclusions

Given the lack of reporting of child sexual abuse, effective strategies are needed to disseminate the General Law on the Rights of Children and Adolescents and the Convention on the Rights of the Child to empower families and communities and promote safe and stigma-free reporting environments. Early interventions should be implemented in key spaces such as schools and family units, addressing issues like strengthening protective factors, developing social–emotional skills, and building secure connections, among other topics, as part of a national policy for preventing child sexual abuse.

## Figures and Tables

**Figure 1 healthcare-13-02489-f001:**
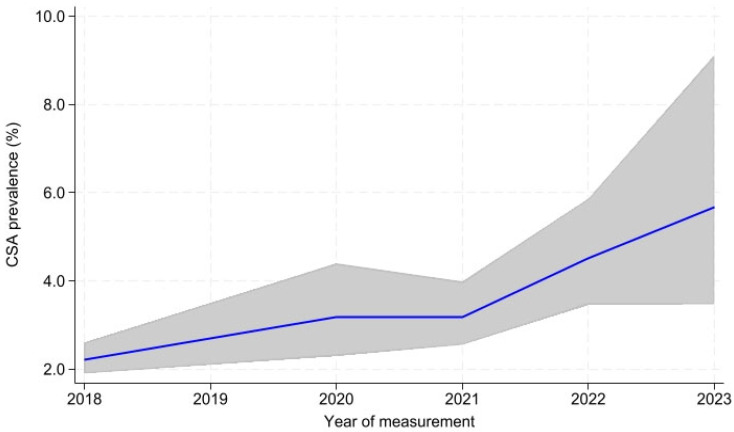
Trends in the prevalence of child sexual abuse (CSA) in Mexico: ENSANUT 2018–2023. ENSANUT: Encuesta Nacional de Salud y Nutrición—National Health and Nutrition Survey.

**Table 1 healthcare-13-02489-t001:** Characteristics of the Study Population, ENSANUT 2018–2023.

		2018	2020	2021	2022	2023
		N ^a^ = 18,818,855	N = 17,460,969	N = 17,720,524	N = 17,899,461	N = 18,187,134
		n ^b^ =14,575	n = 1749	n = 3372	n = 2888	n = 1592
Sociodemographic Variables		%	%	%	%	%
**Sex**						
	Female	49.18	49.13	49.76	49.69	48.02
	Male	50.82	50.87	50.24	50.31	51.98
**Age**						
	10–13	49.86	49.09	50.14	52.08	49.65
	14–17	50.14	50.91	49.86	47.92	50.35
**Residence**						
	Rural	25.76	25.92	24.20	25.04	25.28
	Urban	74.24	74.08	75.80	74.96	74.72
**Socioeconomic Status**						
	Low	34.13	36.81	34.67	34.29	38.18
	Medium	33.77	31.63	33.41	32.84	30.62
	High	32.10	31.55	31.92	32.87	31.20
**Presence of child sexual abuse**						
	No	97.78	96.81	96.81	95.48	94.34
	Yes	2.22	3.19	3.19	4.52	5.66
**Sex of the abuser**						
	Female	11.94	12.64	13.71	5.69	5.07
	Male	88.06	87.36	86.29	94.31	94.93
**Sexual abuser**						
	Partner/boyfriend or girlfriend	1.38	3.04	3.79	4.56	0.21
	Relative	47.03	54.76	43.13	50.27	78.51
	Friend	13.59	11.04	15.13	10.84	5.75
	Neighbor	15.81	18.51	18.67	16.99	12.57
	Stranger	22.19	12.64	19.27	17.34	2.96
**Received care after the aggression**						
	No	72.47	85.71	71.60	82.55	69.38
	Yes	27.53	14.29	28.40	17.45	30.62
**Filed a complaint to the authorities**						
	No	83.20	84.48	81.40	80.48	93.31
	Yes	16.80	15.52	18.60	19.52	6.69
**Reason for failure to file complaint**						
	Fear	34.34	42.05	38.04	48.59	23.79
	Shame	23.40	29.06	16.09	14.98	10.56
	Threats	5.13	0.00	2.30	2.04	10.00
	Did not know they could	17.70	16.69	16.29	18.34	28.36
	Other (specify)	19.43	12.21	27.28	16.05	27.29
**Place of complaint**						
	Public Prosecutor	76.42	60.74	90.97	89.55	91.65
	DIF ^c^	3.44	0.00	0.00	10.45	8.35
	Trustee	2.81	0.00	0.00	0.00	0.00
	Other (specify)	17.33	39.26	9.03	0.00	0.00

ENSANUT: Encuesta Nacional de Salud y Nutrición—National Health and Nutrition Survey. ^a^ Estimated population for 2018 (N = 18,818,855), 2020 (N = 17,460,969), 2021 (N = 17,720,524), 2022 (N = 17,899,461) and 2023 (N = 18,187,134). ^b^ Analytical sample. ^c^ DIF: National System for Integral Family Development.

**Table 2 healthcare-13-02489-t002:** Prevalence of child sexual abuse by sociodemographic characteristics and year of survey: ENSANUT 2018–2023.

	2018		2020		2021		2022		2023	
	%	(95%CI)	%	(95%CI)	%	(95%CI)	%	(95%CI)	%	(95%CI)
**CSA prevalence**	2.22	(1.90–2.60)	3.19	(2.30–4.40)	3.19	(2.55–3.99)	4.52	(3.46–5.86)	5.66	(3.47–9.12)
**Sex**										
Male	1.16	(0.84–1.60)	0.94	(0.41–2.16)	1.39	(0.80–2.39)	2.26	(1.11–4.55)	4.36	(1.57–11.51)
Female	3.32	(2.78–3.97)	5.48	(3.88–7.68)	5.06	(3.99–6.39)	6.79	(5.17–8.88)	7.08	(4.84–10.23)
**Age (years)**										
10–13	1.02	(0.78–1.32)	2.15	(1.24–3.72)	2.02	(1.36–3.00)	3.91	(2.39–6.34)	4.12	(1.76–9.32)
14–17	3.43	(2.85–4.11)	4.19	(2.79–6.25)	4.37	(3.29–5.78)	5.18	(3.88–6.87)	7.18	(3.80–13.16)
**Residence**										
Rural	1.60	(1.14–2.23)	2.36	(1.16–4.77)	2.38	(1.49–3.76)	3.61	(1.62–7.84)	3.87	(1.95–7.53)
Urban	2.44	(2.05–2.91)	3.48	(1–5.01)	3.46	(2.68–4.45)	4.82	(3.70–6.27)	6.27	(3.54–10.84)
**Socioeconomic status**										
Low	2.20	(1.68–2.88)	2.77	(1.46–5.17)	3.27	(2.20–4.85)	3.96	(2.62–5.94)	5.25	(2.30–11.55)
Medium	2.43	(1.90–3.11)	3.06	(1.75–5.32)	3.44	(2.37–4.96)	5.33	(3.20–8.75)	7.77	(3.51–16.35)
High	2.03	(1.51–2.71)	3.80	(2.23–6.40)	2.84	(1.90–4.24)	4.29	(2.72–6.71)	4.08	(1.94–8.37)

ENSANUT: Encuesta Nacional de Salud y Nutrición—National Health and Nutrition Survey, CSA: Child Sexual Abuse, 95%CI: 95% Confidence Interval.

**Table 3 healthcare-13-02489-t003:** Factors associated with child sexual abuse, ENSANUT 2018–2023.

Variables		AOR	*p* Value	95%CI
Year of Survey	2018	1.00		
	2020	1.47	0.042	(1.01–2.12)
	2021	1.45	0.009	(1.10–1.92)
	2022	2.11	0.000	(1.52–2.92)
	2023	2.70	0.000	(1.55–4.68)
Sex				
	Male	1.00		
	Female	2.83	0.000	(1.72–4.68)
Age in years				
	10–13	1.00		
	14–17	1.89	0.003	(1.25–2.86)
Residence				
	Rural	1.00		
	Urban	1.59	0.053	(0.99–2.54)
Socioeconomic status				
	Low	1.00		
	Medium	1.12	0.632	(0.71–1.78)
	High	0.80	0.370	(0.49–1.30)

AOR: Adjusted Odds Ratio, 95%CI: 95% Confidence Interval.

## Data Availability

The raw data supporting the conclusions of this article is available at https://ensanut.insp.mx/index.php (accessed on 29 September 2024).

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
