# Peer review of "Trends in the Prevalence and Case Characteristics of Child Sexual Abuse in Mexico, 2018–2023"

_healthcare, 2025, doi:10.3390/healthcare13192489_

Round 1
Reviewer 1 Report
Comments and Suggestions for Authors
The manuscript aims to determine the trends in the prevalence of child sexual abuse in a representative sample of children in Mexico via the survey: National Health and Nutrition Survey (ENSANUT) for the years 2018, 2020, 2021, 2022 and 2023.
The manuscript is lacking in theory, literature, significance, methods and context. The manuscript needs a critical engagement with the existing literature e.g., Fry, Salter, Finkelhor, the Luxembourg Guidelines etc. A lot of systematic reviews on prevalence have been published e.g. https://doi.org/10.1001/jamapediatrics.2024.5326 and VACS studies Resources | Together for Girls which would ground this study. There needs to be more contextualisation of Mexico e.g. mandatory reporting. Justification is required for the exclusion of certain gender identities; ages of children; perpetrators as children/young people; lifetime vs last year abuse; online and technology-facilitated abuse; exploitation; harmful sexual behaviour/problematic sexual behaviour etc.
How were children and young people who answered the surveys supported? Were there counselling sessions? What is the well-being plan for researchers working in the field of child sexual abuse? As the law in Mexico states: “It is the obligation of any person who has knowledge of cases of children and adolescents who suffer or have suffered, in any way, a violation of their rights, to immediately bring it to the attention of the competent authorities...” Does this mean the the researchers were obligated to report the instances of abuse which were reported in the survey?
Some specific points:
- CSEA is a global issue - why only data from Mexico?
- Informal language e.g "kids", "our country".
- Why is exploitation e.g. CSAM, exortion etc. excluded?
- Line 72 "make sure children are protected, we need up-to-date data on how common CSA is" - how does prevalence data protect children?
- Why only: gender (male and female) and age (10–13 years, 14–17 years)? What about children who are non-binary or below 10?
- “Throughout your life, has anyone ever fondled, touched, or caressed any part of your body or had sex with you when you were very young?” Why are TF-CSEA and exploitation excluded? Does this study only focus on contact abuse? Why?
- The language is problematic "had sex with you when you were very young?" Abuse of a child is never sex.
- What if the perpetrator were another child/young person? “Was the person who did it a man or a woman?” Why are only adult perpetrators included?
- How were data transferred and stored? What are the data protection laws in Mexico?
- Were participants anonymous?
- A proofread is required.
Author Response
"Please see the attachment."

Reviewer 2 Report
Comments and Suggestions for Authors
This manuscript examines trends in the prevalence and characteristics of case of child sexual abuse in Mexico during the period 2018-2023. Child Sexual Abuse (CSA) represents a global problem, one to which Mexico is not immune. This manuscript adds to the cross national literature tracing trends and characteristics of child sexual abuse over time.
Using data on a total of 24,179 children from ages 10-17, drawn from the National Health and Nutrition Survey, the authors trace patterns in child sexual abuse in Mexico during this time period. The data are weighted and representative of the national population. The authors use descriptive statistics to present trends in sexual abuse as well as multiple logistic regression to identify factors associated with CSA in this population.
Measures used were sociodemographic variables (gender, age-dichotomized to 10-13 and 14-17), place of residence (rural vs urban), and SES (high, medium, low). CSA variables used were the dependent variable of CSA experience, measured as Yes before age 12, Yes 12 or older, and No, never. Other CSA measures were relationship to perpetrator, csa treatment (dichotomized to 0=no treatment, 1=treatment), reporting (0=no, 1=yes), reason for not reporting, and reporting location.
Key findings include familiar patterns, such as the majority of the victims being girls, the majority of the abusers being male, being family members, and for three of the five years the victims tended to be older (between 14 and 17). Other findings included a statistically signifcant increase in csa during the five year period studied.
The manuscript has a number of strengths. It adds both to the limited literature on CSA in Mexico and to the international literature on this phenomenon. It uses a nationally representative sample of children over an extended period of time (five years). The manuscript is well written and clearly presented. It is grounded in and cites relevant literature.
With this said, there are some limitations and issues that need to be addressed.
The authors need to say more about the nature of the data set, specifically the nature of the data collection. Were data collected from children via interviews or surveys or both? What was the nature of parental involvement in the data collection process, particularly if interviews were used? If interviews were used, what were the nature and settings for the interviews? How long were the interviews? If surveys were used, what measures were taken to ensure they were age-appropriate?
For example, it strains credulity to think many children would feel comfortable admitting csa in the presence of a parent who may well have been the offender. Hence, readers need to know more about the nature of data collection (survey vs interview), and if it was through an interview the reader needs to know more about the circumstances. For example, were there differences in the nature or form of the data collection for younger (ages 10-13) and older (14-17) populations? Information about the data collection is needed to help readers better understand the reporting process, particularly given the low rates of reporting by the children that were found. The authors should not merely present this low rate without question, particularly when they cite studies that tend to show significantly higher rates of abuse. They need to consider more directly the impact of factors such as forgetting, dissociative amnesia on the part of victims, or fear of reporting victimization by a family member on whom they depend.
The authors report lower prevalence rates of CSA than other studies they cite, such as Stoltenborg et al (2011). The bulk of studies from other societies appear to generally indicate higher rates of CSA than this study. The authors need to address this more directly. Were their findings more accurate than other studies, in that they were able to use a nationally representative sample? Or, is it the case that the data collection methods (perhaps combined with factors related to the victims that are mentioned above) in this survey impacted rates of reporting by the participants? As mentioned, its important that the authors not accept this uncritically. They need to address possible reasons for this significant difference.
The authors should address more fully possible reasons for the increase in CSA noted during the 5 year period. They offer some speculation about social disruptions produced by COVID that may have impacted this but they should address this issue more fully. Since they have identified an interesting trend indicative of a statistically significant increase in reports of CSA by participants this warrants greater attention.
Related to this, data from 2019 were not used. The authors need to identify in the data section why data from this year were not used.
An additional issue relates to reporting cases to authorities. The measure of CSA reporting asks if they or their family reported to authorities. Rates of reporting were very low. The authors should address possible ways to encourage or facilitate reporting to authorities. The most frequently cited reason for not reporting was fear. Cross national literature recognizes the problem of undereporting to authorities. Are there circumstances unique to Mexican society that may have also impacted the likelihood of either victims or family members reporting? Additionally, would younger victims necessarily know whether their victimization was reported?
A final issue relates to future research. A cursory mention (Line 294-296) is made of the need to separate the question used to measure CSA. The authors should identify other avenues that need to be explored, such as reasons for the apparently lower prevalence rate of csa as compared to other studies that they cite that show consistently higher rates of prevalence. Some examples of studies finding higher prevalence rates are the systematic review by Piolanti et al (2025), Barth et al (2013), and Cagney et al (2025). Questions to consider could include whether Mexico is unique, or if/how methodological differences may account for differing findings (lower prevalence rates)of this nature. Either way, further research on the nature, trends, and case characteristics of CSA in Mexico is warranted and the authors should elaborate more fully on possible avenues.
Author Response
"Please see the attachment."
